# A Facile Way to Fabricate GO-EDA/Al_2_O_3_ Tubular Nanofiltration Membranes with Enhanced Desalination Stability via Fine-Tuning the pH of the Membrane-Forming Suspensions

**DOI:** 10.3390/membranes13050536

**Published:** 2023-05-22

**Authors:** Chunxiao Ding, Hong Qi

**Affiliations:** College of Chemical Engineering, Nanjing Tech University, Nanjing 210009, China; 202061104125@njtech.edu.cn

**Keywords:** graphene oxide membrane, Al_2_O_3_ tubular membranes, desalination stability, salt rejection

## Abstract

Pristine graphene oxide (GO)-based membranes have proven promising for molecular and ion separation owing to efficient molecular transport nanochannels, but their separation ability in an aqueous environment is limited by the natural swelling tendency of GO. To obtain a novel membrane with anti-swelling behavior and remarkable desalination capability, we used the Al_2_O_3_ tubular membrane with an average pore size of 20 nm as the substrate and fabricated several GO nanofiltration ceramic membranes with different interlayer structures and surface charges by fine-tuning the pH of the GO-EDA membrane-forming suspension (pH = 7, 9, 11). The resultant membranes could maintain desalination stability, whether immersed in water for 680 h or operated under a high-pressure environment. When the pH of the membrane-forming suspension was 11, the prepared GE-11 membrane showed a rejection of 91.5% (measured at 5 bar) towards 1 mM Na_2_SO_4_ after soaking in water for 680 h. An increase in the transmembrane pressure to 20 bar resulted in an increase in the rejection towards the 1 mM Na_2_SO_4_ solution to 96.3%, and an increase in the permeance to 3.7 L·m^−2^·h^−1^·bar^−1^. The proposed strategy in varying charge repulsion is beneficial to the future development of GO-derived nanofiltration ceramic membranes.

## 1. Introduction

The increasing demand for water resources and the increase in pollution issues call for advanced purification technologies to obtain clean water for drinking from brackish water. Nanofiltration has been widely used for sewage treatment and water purification due to its low cost, high efficiency, and environmental friendliness [1]. Due to the small pore size, traditional nanofiltration ceramic membranes usually require a multi-layer structure on top of a macroporous substrate, and the preparation process is complicated. The higher the membrane selectivity required, the greater the number of support layers needed [2], which is one of the bottlenecks restricting the large-scale preparation and application of nanofiltration ceramic membranes. Therefore, there is an urgent need to look for a new strategy to simplify the fabrication of nanofiltration ceramic membranes.

In recent years, a novel two-dimensional material, graphene, as well as its derivatives, has been one of the research focuses in the area of membrane separation, due to its ultrathin thickness, good thermal and chemical stability, and high ion selectivity [3]. Graphene oxide (GO), for example, as a single atomic layer nanosheet rich in oxygen functional groups, has a large sheet diameter and can be directly stacked into layers to form 2D nanochannels. This advantage endows it with ultrafast water permeability and the ability to hinder salt ion transport [4,5], achieving a size-sieving effect. In addition, the GO nanosheet is negatively charged over a wide range of pH due to deprotonation of the carboxyl group, and the Donnan repulsion theory can be used to explain the primary separation mechanism for its desalination applications [6].

GO-based membranes have shown outstanding performance in gas separation [7,8], and pervaporation [9,10]. However, the practical application of GO membranes for water purification is limited by poor structural stability, poor separation, and difficulty in large-scale manufacturing. Self-supporting GO films, however, are not thick enough to be applied in practical industrial applications. The ceramic membranes have the advantages of good thermal stability, acid and alkali resistance, high strength, and long life [11]. The highly practical value of GO can be achieved by loading on a ceramic substrate to fabricate a GO nanofiltration ceramic membrane.

The GO nanofiltration ceramic membrane is obtained by applying pressure filtration and depositing GO on the substrate membrane. Due to its simple operation, high selectivity of the substrate, and uniform GO deposition, this method is widely used for large-scale preparation of the layered GO-based membrane. For high-pressure and long-term water treatment applications, there are also two intractable problems with GO nanofiltration ceramic membranes: the binding force between GO nanosheets and ceramic substrates and the binding stability between GO nanosheets. The interfacial adhesion between the GO nanosheets and the ceramic substrate is maintained only by hydrogen bonding interactions, and the GO nanosheet is easily peeled off from the ceramic substrate during cross-flow filtration [12]. With the development of mussel biomimetic deposition technology represented by dopamine (PDA), chemical modification with PDA has become an effective method to improve composite membranes’ binding force and stability [13]. It could polymerize and stick on all kinds of organic and inorganic surfaces through the formation of strong covalent and noncovalent bonds with surfaces [14]. In addition, the unmodified GO nanosheets only contain hydrogen bonds and π–π interactions [15], and plenty of oxygen-containing functional groups intensely hydrated with water molecules, resulting in additional electrostatic repulsion. Weak hydrogen bonds or intermolecular interactions are insufficient to connect adjacent GO nanosheets in a stable way, causing poor swelling, redispersion, and exfoliation of GO membranes in an aqueous solution. To improve the interlayer stability of GO nanosheets, the diamine monomer molecule ethylenediamine (EDA) has been widely used to bond adjacent GO nanosheets [16,17,18]. The layer spacing between GO nanosheets is regulated by the covalent cross-linking of amine groups in EDA with GO carboxyl groups and epoxy groups through nucleophilic addition. However, these methods fail to gain small enough interlayer nanochannels to achieve high salt rejection. Apart from interlayer nanochannels, oxygen-containing groups of GO are also crucial for salt rejection, since these functional groups on GO could provide electrostatic interactions between GO nanosheets and ions. The negatively charged surface of GO membranes plays a vital role in transporting ions due to the charge selectivity mechanism [19,20,21].

To improve the desalination capability of GO nanofiltration ceramic membrane, based on the consideration of changing the charge of the GO membrane, we simplified the preparation of nanofiltration ceramic membranes by fine-tuning the pH of the membrane-forming suspensions. We used a PDA-modified ceramic substrate to enhance the interfacial bonding ability between GO nanosheets and the substrate. Meanwhile, EDA was chosen as a cross-linking agent to enhance the binding force between GO nanosheets and the stability of GO nanofiltration ceramic membranes in aqueous solutions. The resulting membranes do not require an excessive multi-layer structure and are superior to the conventional nanofiltration membranes with a complex preparation process. Moreover, unlike in other studies, we studied the long-term stability of the GO nanofiltration ceramic membrane in the high-pressure desalination process. In addition to excellent long-term stability, the resulting GO nanofiltration ceramic membrane also exhibits improved desalination performance.

## 2. Materials and Methods

### 2.1. Materials

GO suspension (1 mg/mL) was supplied by Xianfeng Nanomaterials Technology Co., Ltd., Nanjing, China. Tris (hydroxymethyl) aminomethane and Dopamine hydrochloride (PDA, AR, 99.9%) were obtained from Aladdin Biochemical Technology Co., Ltd., Shanghai, China. Anhydrous ethylenediamine (EDA, AR, 99%), hydrochloric acid (65 wt%), NaOH, Na_2_SO_4_, NaCl, MgSO_4_, and MgCl_2_ were received from Shanghai Lingfeng Chemical Reagent Co., Ltd., Shanghai, China. The tubular Al_2_O_3_ ceramic membrane (pore size: 20 nm, outside diameter: 12 mm, inside diameter: 8 mm, length: 110 mm) used as support was obtained from Nanjing Hongyi Ceramic Nanofiltration Membranes Co., Ltd., Nanjing, China.

### 2.2. Preparation of PDA-Modified Tubular Al_2_O_3_ Ceramic Membrane

First, a certain amount of Tris was added to deionized (DI) water and under stirring. Then, the pH of the Tris–HCl solution was adjusted to 8.5 with 1 mM hydrochloric acid to form a 5 mM Tris–HCl buffer. Dopamine (1 mg/mL) was then dissolved in 5 mM Tris–HCl solution (pH: 8.5) in an Erlenmeyer flask. After the solution was thoroughly mixed, the Al_2_O_3_ tubular membrane was soaked in the above-mentioned solution for 20 h to allow a self-polymerization of PDA under a shaded environment. After that, the membrane was flushed with DI water to get rid of the residual dopamine on the surface. Finally, the above membrane was dried at 60 °C for 2 h to obtain a PDA-modified tubular Al_2_O_3_ membrane (hereafter referred to as PDA-Al_2_O_3_ membrane).

### 2.3. Preparation of EDA-Crosslinked GO-EDA/Al_2_O_3_ Membrane

First, 1 mg/mL GO solution was diluted with deionized water, and 200 mL of GO coating solution was uniformly dispersed after ultrasonication at 35 kHz for 5 min. Next, 1 mM EDA solution was put into the mixture and stirred at 80 °C for 1 h to obtain GO-EDA membrane-forming suspension. The pH of the pristine membrane-forming suspension was 10.04. After the membrane-forming suspension was cooled, HCl or NaOH (1 M, 0.1 M) were used to regulate the pH of the membrane-forming suspension to 7.00, 9.00, and 11.00, respectively. The resulting membrane-forming suspension was deposited on the inner surface of the PDA-Al_2_O_3_ membrane to form the separation layer driven by pressured nitrogen at 1 bar using a self-designed filtration device. Finally, freshly prepared membranes were dried at 40 °C for 12 h to obtain GO-EDA/Al_2_O_3_ membranes, labeled as GE-P, GE-7, GE-9, and GE-11, respectively. The scheme of the GO-EDA/Al_2_O_3_ membranes preparation is shown in Figure 1.

### 2.4. Characterizations

The microscopic morphology of GO-EDA/Al_2_O_3_ membranes was characterized by a scanning electron microscope with electron dispersive spectrometry (SEM-EDS, S-4800, Hitachi, Tokyo, Japan). Surface charge data of the GO-EDA/Al_2_O_3_ membrane were measured using a zeta-sizer instrument (DLS, ZS-90, Malvern Panalytical Ltd., Malvern, UK). Water contact angle (CA) of the GO-EDA/Al_2_O_3_ membrane was were obtained using a contact angle tester (Drop Meter A-100P, Haishu Maishi Scientific Test Co. Ningbo, China). X-ray photoelectron spectroscopy (XPS, ESCALAB 250, Thermo Scientific, Waltham, MA, USA) was employed to characterize the surface compositions of the GO membranes. Determination of the conductivity of ions in salt solutions was performed using a conductivity meter (DDS-307A, INESA Scientific Instrument Co., Ltd., Shanghai, China).

### 2.5. Evaluation of Membrane Performance

The permeability and separation performance of as-prepared membranes were tested using 1 mM salt solution on a homemade cross-flow filtration system (Figure 2) at 20 °C. The transmembrane pressure was 5 bar, and the flow rate was 1.3 m·s^−1^. The effective membrane area for separation was 2.3 × 10^−3^ m^2^. The stability of the GO-EDA/Al_2_O_3_ membrane was evaluated according to an experiment of long-term immersion in water.

After soaking the GO-EDA/Al_2_O_3_ membrane in pure water for 170 h, 340 h, 510 h, and 680 h, respectively, the permeance of pure water and the rejection of the above four salt solutions were measured under the corresponding soaking time. An electrical conductivity meter was used to directly test the salt concentration. The pure water permeance *J* (L·m^−2^·h^−1^·bar^−1^) and the salt rejection *R* are calculated as the following equations:(1)$$J=\frac{V}{At\Delta P}$$
(2)$$R=(1-\frac{C_{p}}{C_{f}})\times 100\%$$
where *V* (L) denotes the volume of permeate; *A* (m^2^) means the effective area of the membrane; *t* (h) is permeation time; Δ*P* (bar) is the operating pressure; and *C_f_* and *C_p_* represent the feed and permeate concentrations of salt solutions, respectively.

## 3. Results and Discussion

### 3.1. Characterization and Permeability of PDA-Al_2_O_3_ Membrane

The surface morphology of the Al_2_O_3_ membrane and PDA-Al_2_O_3_ membrane are characterized by SEM. It was observed on the optical photos that the color of the membrane surface turned dark after PDA modification. As observed from Figure 3c, the alumina particles were uniformly distributed on the surface of the Al_2_O_3_ membrane, and an amount of small valley-ridge structures can be clearly found on the top surface of the PDA-Al_2_O_3_ membrane. From the EDS mapping images in Figure 3e1–e5, the N element from PDA was introduced and evenly distributed on the membrane, indicating the formation of a PDA layer on the Al_2_O_3_ membrane surface. The pure water permeance of the Al_2_O_3_ and PDA-Al_2_O_3_ membranes was tested. It can be seen from Figure 4 that the pure water permeance (221.6 L·m^−2^·h^−1^·bar^−1^) of the PDA-Al_2_O_3_ membranes is reduced by 11.4% compared with the Al_2_O_3_ membrane. In combination with the characterization analysis and the decrease in membrane flux after PDA treatment, we concluded that catechol in PDA underwent oxidation and polymerization through deprotonation and intermolecular Michael addition reaction under an aerobic and alkaline condition, and formed crosslinked homopolymer grafted on the surface of Al_2_O_3_ membrane [22,23].

### 3.2. Characterizations of GO-EDA/Al_2_O_3_ Membranes

The GO-EDA membrane-forming suspensions with different pH values were prepared and analyzed below. It is evident from Figure 5a that the average size of GO nanosheets was smaller when the membrane-forming suspension was in a neutral environment as compared to that of an alkaline condition. The most chemically reactive part of GO is the acidic carboxyl group, and the sheet size depends on the carboxyl groups at the edges of GO nanosheets [24]. Thus, the different agglomeration states of GO nanosheets can lead to differences in size under different acid-based environments.

To analyze the surface charges of the GO-EDA/Al_2_O_3_ membranes, different pH values were adjusted to the GO-EDA suspensions. The zeta potential was tested after the suspension system was stabilized. The zeta potential measured at different pH values showed that the GO-EDA membrane-forming suspensions were always negatively charged at the four pH values, mainly owing to the proton effect in carboxyl groups. According to Figure 5b, the absolute value of zeta potential increases with the increase in pH value, probably because the rise in pH value causes a decrease in H^+^ concentration in the solution. The carboxyl groups on GO-EDA are more likely to be ionized, which increases the content of negative charges [25]. The hydroxyl groups form hydrated anions with the abundant OH^−^ in the solution, which further increases the negative charge contained in the structural layer.

In order to further study the chemical composition of the membrane surface, XPS was used to analyze the four GO-EDA/Al_2_O_3_ membranes. The results are shown in Figure 6. It can be seen from Figure 6 that the characteristic peaks of N 1s appeared on the surface of the four membranes, which indicated the successful crosslinking of EDA.

The microstructures of GO-EDA/Al_2_O_3_ membranes were analyzed with SEM. As shown in Figure 7a1–a4, the GO was deposited on the Al_2_O_3_ substrate, forming a complete and defect-free dense separation layer [26]. As depicted in Figure 7b1–b4, the separation layers were tightly bonded to the Al_2_O_3_ substrate, all the GO-EDA/Al_2_O_3_ membranes exhibited a uniformly laminated structure, and no observable defects can be found in the membrane. It is often observed that by increasing the pH from 7 to 11, the thickness of the GO-EDA/Al_2_O_3_ membrane increased from 101 to 273 nm. For GE-11, the larger membrane thickness is due to the addition of a large amount of NaOH in the membrane-forming suspension, which increased the repulsion between GO nanosheets and the surface adsorption of Na^+^ [27]. In addition, the SEM cross-section images reveal that the thicknesses of GO-EDA/Al_2_O_3_ membranes are several hundred nanometers, which is beneficial to reduce the resistance of mass transfer and enhance water permeance. Figure 7c1–c4 shows the water contact angle of GE-7, GE-9, GE-P, and GE-11, which all exhibit a certain level of hydrophilicity. The abundant functional groups on the GO surface endow it with good hydrophilicity and high chemical activity. The oxygen-containing functional groups on the GO surface will undergo a deoxygenation reaction at a high pH value, resulting in decreased hydrophilicity [28]. When the pH of the membrane-forming suspension is 11, several oxygen-containing functional groups on the GO nanosheets will undergo a reduction reaction and be removed, and the hydrophilicity of the membrane decreases, which is consistent with the decreasing oxygen content of GE-11 in Figure 6.

### 3.3. Pure Water Permeability of GO-EDA/Al_2_O_3_ Membranes

The pure water permeance of GO-EDA/Al_2_O_3_ membranes was tested for a constant time using a laboratory-made tubular membrane cross-flow filtration device. With the increase in the cross-flow filtration time, the pure water permeance decreased at first and then leveled off, and it remained stable for a short time (ranging from 30 min to 60 min), as displayed in Figure 8. The steady-state pure water permeance of GE-7, GE-P, GE-11, and GE-9 are 2.0, 1.5, 1.2, and 1.0 L·m^−2^·h^−1^·bar^−1^, respectively. It may be explained as follows: the high permeance at the early stage of the filtration may be caused by the loose microstructure between the GO nanosheets, and the incomplete stacking of GO nanosheets creates a wide transport channel for water to pass through. However, with the increase in the cross-flow time, the loose microstructure between the GO nanosheets is continuously compacted under the action of pressure, and the additional water transport channels are reduced, resulting in a decrease in permeance [29]. In addition, due to the high aspect ratio of GO nanosheets, the transport path of water through the GO-EDA/Al_2_O_3_ membrane is extended, which results in a lower steady-state water permeance. At the same time, it can be found that compared with the GE-P, the steady-state permeance of the GE-7 reaches a maximum of 2.0 L·m^−2^·h^−1^·bar^−1^. This is due to the protonation of carboxyl groups after the addition of a large amount of acid. As a result, the sheet size is reduced, and the water transport path through the GO-EDA/Al_2_O_3_ membrane with a small-sized GO nanosheet is shorter, thereby increasing its permeance [30].

### 3.4. Desalination Performance of GO-EDA/Al_2_O_3_ Membranes

Figure 9 shows that the GO-EDA/Al_2_O_3_ membranes exhibited distinct rejection toward different salts, which follows the sequence of R(Na_2_SO_4_) > R(MgSO_4_) > R(NaCl) > R(MgCl_2_). The separation behavior can be explained by size sieving and the Donnan exclusion theory [31]. The negatively charged GO-EDA/Al_2_O_3_ membranes are more inclined to repel negatively charged anions while attracting positively charged cations to maintain the charge neutrality of the feed solution. Donnan exclusion theory can be expressed as Equation (3):(3)$$R=1-\frac{C_{B}^{m}}{C_{B}}=1-(\frac{\left|Z_{B}\right|C_{B}}{\left|Z_{B}\right|C_{B}^{m}+C_{X}^{m}}{{{)}^{\left|Z_{B}\right|}}^{/}}^{\left|Z_{A}\right|}$$
where *R* is the salt rejection; *Z_B_* and *Z_A_* are the valences of the co-ions (negatively charged ions) and counterions (positively charged ions); $C_{B}$ and $C_{B}^{m}$ denote the concentration of the co-ions in the solution phase and membrane phase; and $C_{X}^{m}$ represents the charge concentration on the membrane surface. Theoretically, the salt rejection should be determined as R(Na_2_SO_4_) > R(NaCl) ≈ R(MgSO_4_) > R(MgCl_2_). However, the rejection of MgSO_4_ was higher than NaCl in the experiment, which was due to the effect of size sieving. Compared with the hydration radius of Na^+^ and Cl^−^, large-sized hydrated Mg^2+^ and SO_4_^2−^ were more easily blocked by GO nanochannels [1].

In addition, the membranes exhibit a higher rejection after pH adjustment to the membrane-forming suspensions with acid or alkali compared to the GE-P. As the pH increased from 7 to 11, the salt permeance of Na_2_SO_4_ decreased from 1.5 to 0.9 L·m^−2^·h^−1^·bar^−1^, while the salt rejection increased from 67.6% to 86.6%. In this case, the GE-11 shows an excellent salt rejection performance. Regarding salt rejection efficiency, the surface charge interacts with salts through the GO membrane, and the Donnan exclusion theory plays a major role in salt rejection. From the above analysis, we proposed a separation mechanism of GO-EDA/Al_2_O_3_ membranes in Figure 1. More specifically, the alteration of membrane-forming suspension resulted in their different surface charge repulsion and differently sized GO nanosheets. As a result, the enhanced repulsion interaction with salt ions achieved higher salt rejection. Moreover, the GO nanosheets with larger size were inclined to form longer transportation nanochannels in GE-11, which was beneficial to improve the desalination performance.

### 3.5. Stability of GO-EDA/Al_2_O_3_ Membranes

In order to study the desalination stability of GO-EDA/Al_2_O_3_ membranes, experiments at variable immersion time and operating pressures are performed, as illustrated in Figure 10 and Figure 11. With the prolongation of soaking time, the permeance of the four different GO-EDA/Al_2_O_3_ membranes showed a slight upward trend, and so did the rejection towards the four salt solutions. After soaking in water for 680 h, the rejections of Na_2_SO_4_ by four membranes (GE-7, GE-9, GE-P, and GE-11) were 82.1%, 88.6%, 82.0%, and 91.5%, respectively. Among them, the salt rejections of the GE-11 membrane fluctuated within a certain range, and basically it did not change with the prolongation of soaking time, indicating that it has better stability in an aqueous solution. This may result from the residual salt ions after multiple tests of the GO-EDA/Al_2_O_3_ membrane in the salt solution. These residual salt ions produced a cross-linking effect between the GO nanosheets [32], making the membrane layer dense. The thicker the membrane, the fewer the internal defects; the higher the degree of densification, the more pronounced the cross-linking effect.

In addition, we tested the desalination stability of the GE-11 under high-pressure operation after soaking in water for 680 h. Figure 11 shows that both salt flux and Na_2_SO_4_ rejection of GE-11 increase with operating pressure. At 20 bar, the salt flux is 74 L·m^−2^·h^−1^, and the rejection of Na_2_SO_4_ is up to 96.3%. The flux of the GE-11 membrane initially increases slowly with the increase in operating pressure, and when it exceeds a certain pressure, the flux of the GE-11 membrane increases with the increased rate of pressure, and the relationship is basically linear. This is because when the nanofiltration membrane filters salt solution, it needs to overcome the osmotic pressure of the salt solution. When the pressure is lower than the osmotic pressure, the membrane flux is low, and when the pressure exceeds the osmotic pressure, the water in the solution can overcome the osmotic pressure barrier [33,34,35]. While the salt rejection depends on salt concentration in the permeate, when the growth rate of the salt permeance is greater than that of the water permeance, there is a decrease in the salt rejection, and conversely, an increase.

Compared with the properties of the nanofiltration membranes reported in other studies, the GO-EDA/Al_2_O_3_ membrane fabricated in this work showed a competitive desalination performance and permeability, as detailed in Table 1. The higher Na_2_SO_4_ rejection and sufficient permeability of this GO-EDA/Al_2_O_3_ membrane can be ascribed to the optimized membrane-forming suspension and enhanced surface charge.

## 4. Conclusions

In summary, the facile fabrication of the GO-EDA/Al_2_O_3_ membrane was developed by fine-tuning the pH of the GO-EDA membrane-forming suspension. GO nanosheets show different sizes and interlayer charges in membrane-forming suspension with different pH. The resultant membranes could maintain remarkable performance towards desalination stability whether immersed in water for 680 h or placed in a high-pressure environment. With the prolongation of cross-flow filtration time, the pure water permeance of GO-EDA/Al_2_O_3_ membranes showed a trend of first decreasing slightly and then stabilizing. The optimized membrane GE-11, with a higher pH value, showed superior anti-swelling properties and desalination performance. After soaking in water for 680 h, this membrane exhibited a 96.3% rejection of Na_2_SO_4_ and water permeance of 3.7 L·m^−2^·h^−1^·bar^−1^ at 20 bar with a 1 mM Na_2_SO_4_ feed solution. The membrane-forming suspension with higher pH could enhance Donnan repulsion and achieve an effective salt rejection. This work demonstrates a simple approach to fabricating stable GO nanofiltration ceramic membranes and may be helpful in further improving their desalination stability.

## Figures and Tables

**Figure 1 membranes-13-00536-f001:**
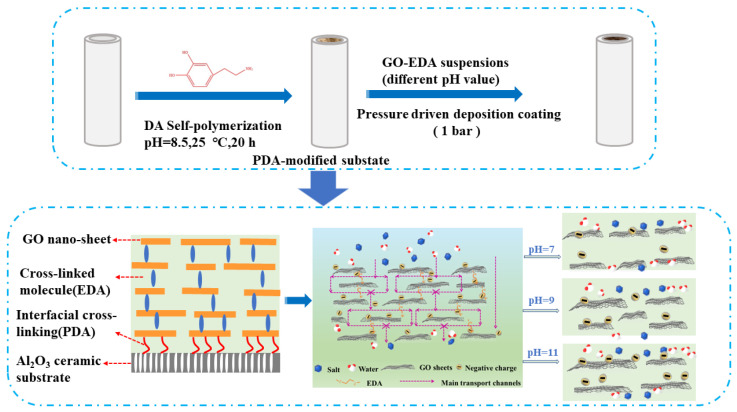
Schematic diagram of fabrication and separation mechanism of GO-EDA/Al_2_O_3_ membranes.

**Figure 2 membranes-13-00536-f002:**
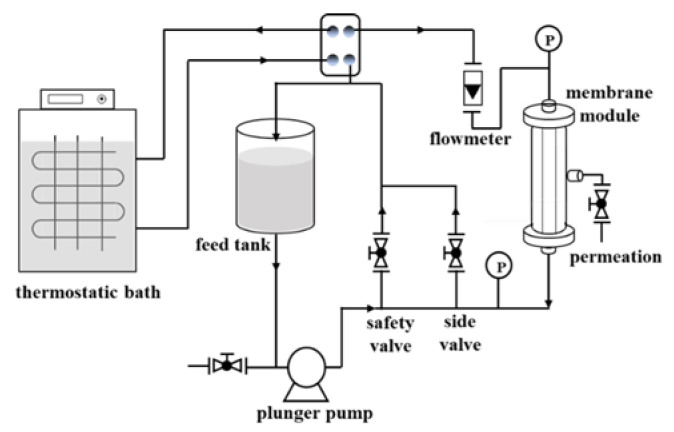
Schematic diagram of cross-flow filtration device.

**Figure 3 membranes-13-00536-f003:**
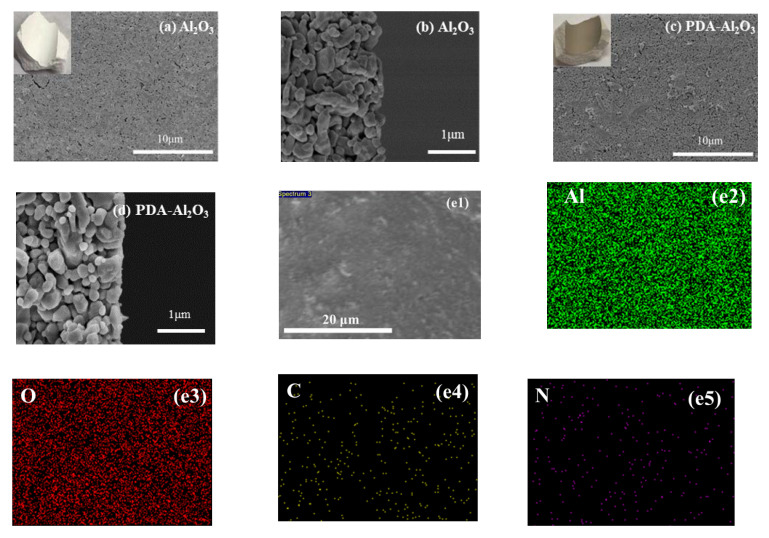
SEM images of top surface of (**a**) Al_2_O_3_ and (**c**) PDA-Al_2_O_3_ membranes, cross-section of (**b**) Al_2_O_3_ and (**d**) PDA-Al_2_O_3_ membranes, and (**e1**–**e5**) EDS mapping of PDA-Al_2_O_3_ membrane.

**Figure 4 membranes-13-00536-f004:**
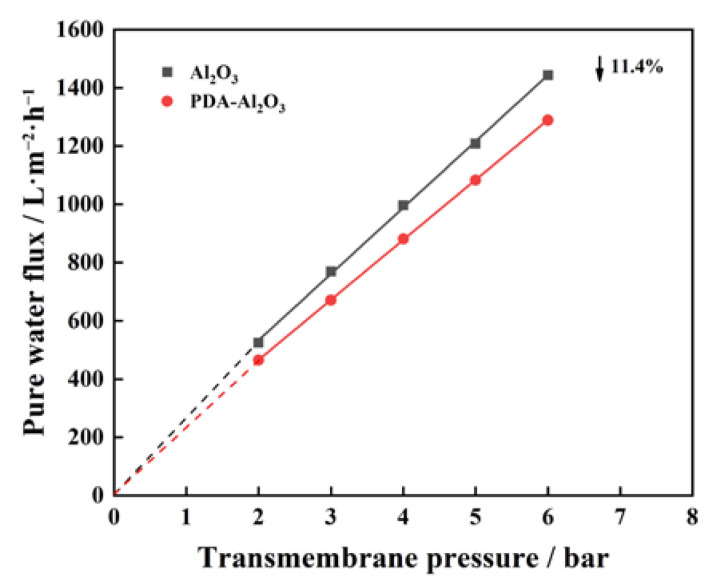
Pure water flux of Al_2_O_3_ and PDA-Al_2_O_3_ membranes.

**Figure 5 membranes-13-00536-f005:**
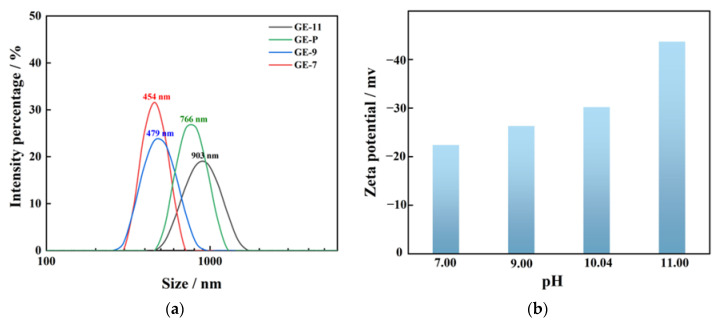
(**a**) Particle size distribution of GO-EDA suspensions; (**b**) Zeta potentials of GO-EDA suspensions.

**Figure 6 membranes-13-00536-f006:**
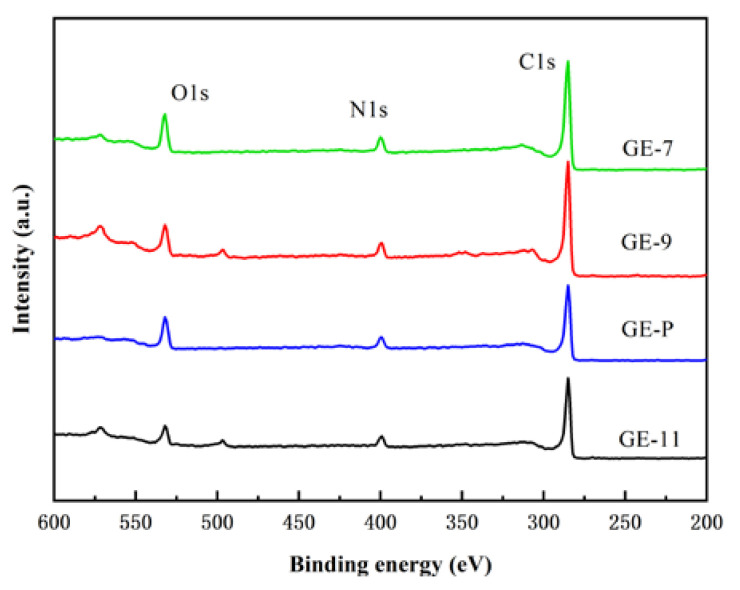
XPS spectra of GO-EDA/Al_2_O_3_ membranes.

**Figure 7 membranes-13-00536-f007:**
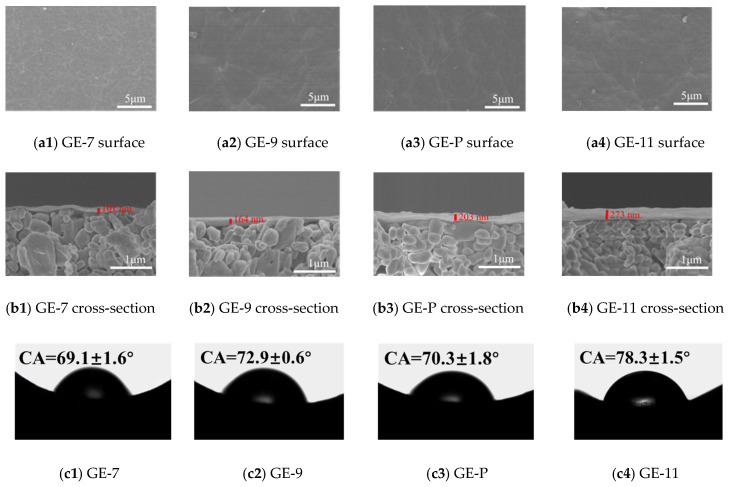
SEM images and CA of GO-EDA/Al_2_O_3_ membranes with a different pH of membrane-forming suspensions.

**Figure 8 membranes-13-00536-f008:**
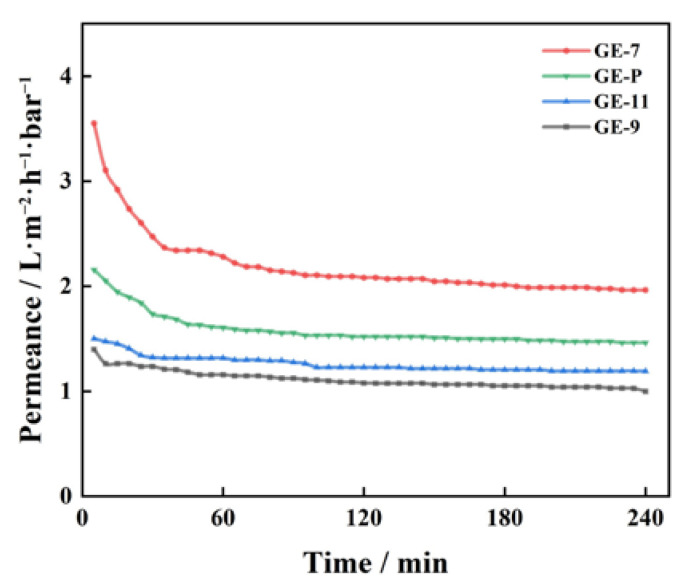
Pure water permeance of GO-EDA/Al_2_O_3_ membrane.

**Figure 9 membranes-13-00536-f009:**
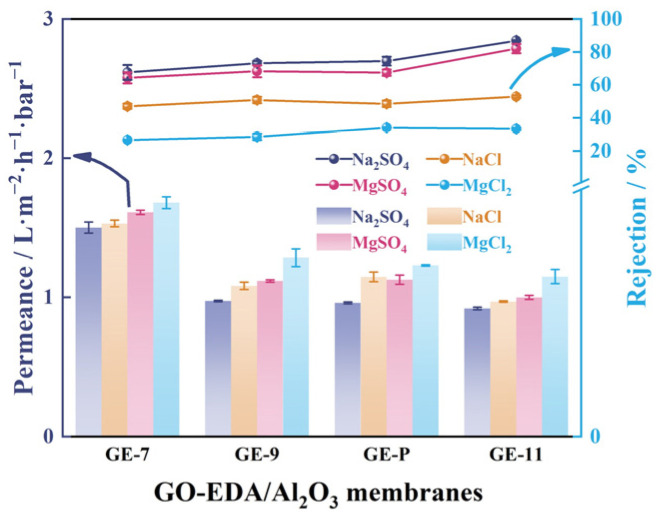
Rejection of GO-EDA/Al_2_O_3_ membranes towards four salt solutions. Error bars represent standard deviations for three measurements.

**Figure 10 membranes-13-00536-f010:**
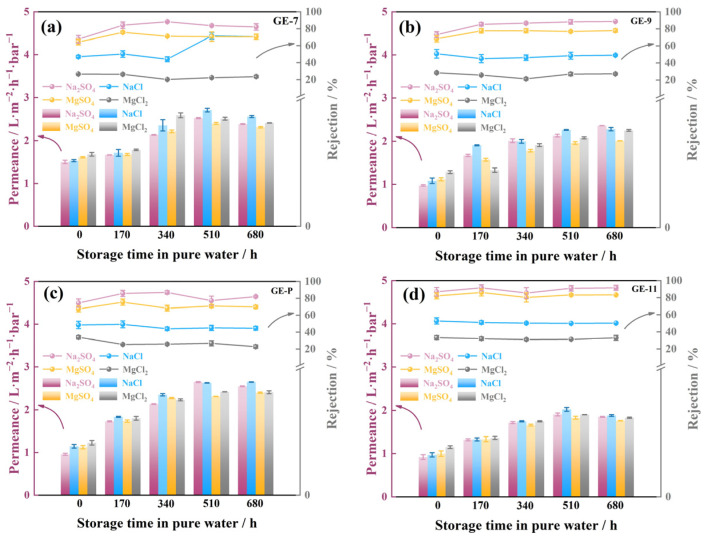
The permeance and rejection stability of GO-EDA/Al_2_O_3_ membranes (**a**) GE-7, (**b**) GE-9, (**c**) GE-P, and (**d**) GE-11. Error bars represent standard deviations for three measurements.

**Figure 11 membranes-13-00536-f011:**
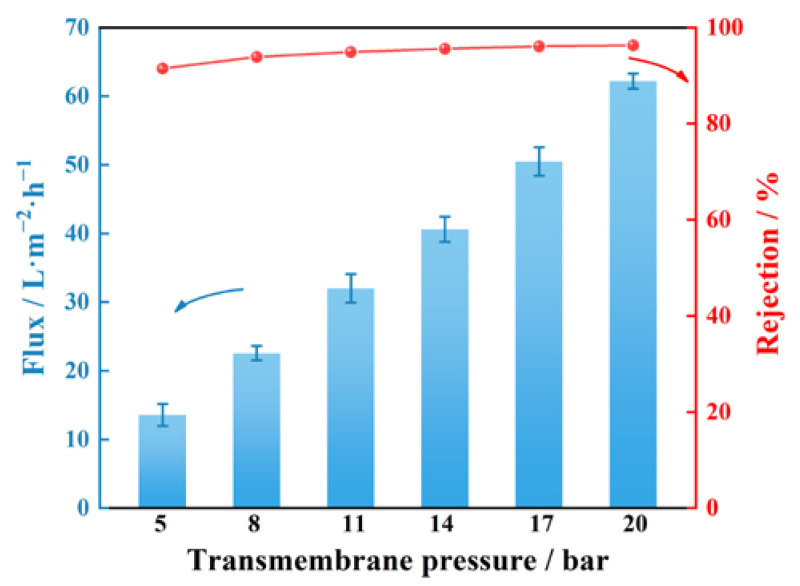
Rejection of GE-11 membrane towards 1 mM Na_2_SO_4_ solution measured under different transmembrane pressures. Error bars represent standard deviations for three measurements.

**Table 1 membranes-13-00536-t001:** Comparison of desalination performance with other GO and commercial nanofiltration membranes.

Membrane	Substrate	Feed Condition	Testing Condition	Water Permeance (L·m^−2^·h^−1^·bar^−1^)	Rejection (%)	Ref.
GO-UR	MCE	NaCl	50 mM, 1 bar	/	25.74	[36]
RGO	PVDF	Na_2_SO_4_, NaCl, MgSO_4_, MgCl_2_	20 mM, 5 bar	3.3	~60, 30, 20, 40	[11]
Pristine GO	α-Al_2_O_3_	Na_2_SO_4_, MgSO_4_MgCl_2_, NaCl	1 mM, 5 bar	3.68	72.6, 58.4, 23.7, 45.8	[37]
GO	Al_2_O_3_	NaCl, MgSO_4_	10 mM, 4 bar	1.25	28.66, 43.52	[38]
GO-PEI	PAN	MgCl_2_	1000 ppm, 5 bar	4.2	86	[24]
GO-PEI	PAN	MgCl_2_, Na_2_SO_4_, NaCl	10 mM, 3.4 bar	4.1	72, 30, 20	[39]
EDA/GO	BPPO	Na_2_SO_4_, MgSO_4_, NaCl	1000 ppm, 1 bar	4.1	56.2, 48, 36.3	[40]
Commercial NF1	PS	Na_2_SO_4_, NaCl	20 mM, 20 bar	3.45	98, 51	[41]
Osmonics CK	/	NaCl	1.5 mM~100 mM	2.42	45.5~77.7	[42]
Osmonics DK	/	NaCl	1.5 mM~100 mM	3.05	22.0~75.6	[42]
Commercial NF2	PAM	Na_2_SO_4_, NaCl	20 mM, 20 bar	6.5	99, 44	[41]
rGO	Al_2_O_3_	/	/, 15 bar	1.7	/	[43]
GO	PSf	Na_2_SO_4_	2000 ppm, 15 bar	11	65	[44]
TiO_2_-GO	Al_2_O_3_	Na_2_SO_4_	1.4 mM, 5 bar	5.6	9.8	[45]
GO/PAH	PAN	Na_2_SO_4_	6.7 mM, 6.9 bar	2	68	[46]
GO-EDA	Al_2_O_3_	Na_2_SO_4_	1 mM, 20 bar	3.7	96.3	This work

## Data Availability

The data presented in this study are available on request from the corresponding authors.

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
