# Peer review of "A Facile Way to Fabricate GO-EDA/Al2O3 Tubular Nanofiltration Membranes with Enhanced Desalination Stability via Fine-Tuning the pH of the Membrane-Forming Suspensions"

_membranes, 2023, doi:10.3390/membranes13050536_

Round 1

Reviewer 1 Report

The article titled "facile way to fabricate GO-EDA/Al2O3 tubular nanofiltration membranes with enhanced desalination stability via fine-tuning the pH of the membrane-forming suspensions" is presented for possible publication in the journal of Membranes.

They proposed a GO-based ceramic membrane on the alumina nanotube as substrate. They also proposed high durability under pressure and severe pH with a salt rejection of higher than 90%. There are some revisions that should be performed on the current manuscript.

1- The literature review in the introduction part is not in-depth enough, especially about GO-based membranes. This is just a rewrite of other's studies that needs more analysis, cons and pros and...

2- In part 2.2: "...into which 1 mg/ml PDA was added..." is not clear. Does it mean 1mg PDA in 1 ml of buffer? It seems not reasonable for alumina ceramic membrane preparation.

3- In part 2.3: How you could have 1M of acid/base to prepare pH of 11 for instance. All samples are from pH 7 and higher. 1M HCL then?

4- The characterization parts should not be in the performance section.

5- In part 3.1: "PDA is closely covered on the Al2O3 membrane without an obvious boundary." How could you conclude this from the SEM image from the surface? It needs other characterization proof on this like EDX. There is no obvious illustration of chemical bonds or correct coating of PDA on alumina.

6- "The results indicated PDA was successfully grafted on the Al2O3 membrane via self-polymerization". Is there any analysis on the washing or stability of PDA graft with alumina ceramic? It seems there is a critical need of studying the rejection-washing-stability of PDA after a time of working under pressure.

6- Figure 4. How can you use the water flux decrease in a beneficial target as the purpose should be higher water flux?

7- 3.2. The physical properties of membranes are shown in a separation section while the first section is obviously a characterization of membrane physics.

8- In Fig.5, there are a bit differences in the size distribution trend. Were the results repeated? If yes, what is the median? The size distribution cannot be fitted with the trend of the sample's specification.

9- There is not enough scientific evidence about Figure 7, for instance, contact angle. The thickness and SEM images cannot confirm the previous images.

10- The water permeability showed a decrease that is not light after the first sample. How can you confirm the differences between samples 7-P and 11-9?

11- The trend of rejection also has a problem. I am wondering if the test were repeated and reported based on an affordable median.

12- "At 20 bar, the salt flux is 74 L·m-2·h-1, and the rejection of Na2SO4 is up to 96.3%.". Do you think that the ceramic alumina can tolerate the 20 bar pressure for the rejection test? If yes, at what temperature? How fragile is the membrane? It seems there is a great lack of mechanical analysis and characterization on this part.

13- Figure 11 is repeatable? The trend is nice but is not match with other characterization parts. Please clarify more about the results of this.

14- Based on the Table1, lots of operation conditions have pressure under 5 bar just one polysulfone substrate could tolerate 20 Bar which did not have other conditions presented.

15- The references can be more relevant and in-depth about the characterizations especially for alumina and GO-based membranes with different aspects of membranes.

Reviewer 2 Report

Thanks for submitting the manuscript to MDPI Membranes; the manuscript required extensive corrections in language and continuity. I suggest improving the overall manuscript structure in-terms of continuity and scientific discussion.  Specific comments are;   1. Please elaborate on what you mean by Pure GO, I don't think it is appropriate to use Pure GO. GO means GO. 2. The first two paragraphs are very general and are not required; authors may rewrite them in 2-3 lines max for general information. 3. Add some technical data and discussion. How is your method, material, or approach different from the existing one? 4. Authors are requested to provide additional FE-SEM images for a better understanding of morphology. 5. Too much variation is observed in WCA, Please re-do the analysis and provide fresh images of WCA. 6. Analysis graphs can be improved. 7. Please add relevant comparison studies to Table 1. and add a few more.   Overall, the authors are urged to revise the manuscript for another round of review. A minor revision is suggested for the present manuscript.  Thanks for submitting the manuscript to MDPI Membranes; the manuscript required extensive corrections in language and continuity. I suggest improving the overall manuscript structure in-terms of continuity and scientific discussion.  Specific comments are;   1. Please elaborate on what you mean by Pure GO, I don't think it is appropriate to use Pure GO. GO means GO. 2. The first two paragraphs are very general and are not required; authors may rewrite them in 2-3 lines max for general information. 3. Add some technical data and discussion. How is your method, material, or approach different from the existing one? 4. Authors are requested to provide additional FE-SEM images for a better understanding of morphology. 5. Too much variation is observed in WCA, Please re-do the analysis and provide fresh images of WCA. 6. Analysis graphs can be improved. 7. Please add relevant comparison studies to Table 1. and add a few more.   Overall, the authors are urged to revise the manuscript for another round of review. A minor revision is suggested for the present manuscript. 
